# Determination of Micromovements in Removable Prosthesis during Mastication: A Pilot Study with 3D Electromagnetic Articulography

**DOI:** 10.3390/bioengineering11030229

**Published:** 2024-02-28

**Authors:** Franco Marinelli, Camila Venegas, Joaquin Victorio Ruiz, Nicole Farfán-Beltrán, Erwin Staub, Pablo Navarro, Josefa Alarcón-Apablaza, Ramón Fuentes

**Affiliations:** 1Research Centre in Dental Sciences (CICO-UFRO), Dental School, Facultad de Odontología, Universidad de La Frontera, Temuco 4780000, Chile; franco.marinelli@ufrontera.cl (F.M.); camilabelen.venegas@ufrontera.cl (C.V.);; 2Institute for Research and Development in Bioengineering and Bioinformatics (IBB), CONICET-UNER, Oro Verde E3100, Argentina; joaquin.ruiz@uner.edu.ar; 3Master Program in Dentistry, Dental School, Universidad de La Frontera, Temuco 4780000, Chile; 4Facultad de Ciencias de la Salud, Universidad Autónoma de Chile, Temuco 4780000, Chile; 5Doctoral Program in Morphological Sciences, Faculty of Medicine, Universidad de La Frontera, Temuco 4780000, Chile; 6Department of Integral Adults Dentistry, Dental School, Universidad de La Frontera, Temuco 4780000, Chile; 7Scientific and Technological Bioresource Nucleus (BIOREN-UFRO), Universidad de La Frontera, Temuco 4780000, Chile

**Keywords:** edentulism, removable prostheses, micromovements, electromagnetic articulography, mastication, chewing

## Abstract

Edentulism can generate negative impacts on self-esteem, interpersonal relationships, and oral functions. Removable prostheses are commonly used for tooth replacement, but they may cause discomfort due to micromovements during mastication. Objective and quantifiable methods are needed to evaluate these micromovements. A pilot study was conducted to determine the micromovements in removable prostheses during mastication using a 3D electromagnetic articulography (EMA-3D) system. One elderly participant wearing lower removable prostheses and an upper total well-fitting removable prosthesis was studied. The EMA-3D system was used to record movements in three spatial planes. Peanuts were given as test food, and the participant was instructed to chew normally while recordings were carried out until feeling the need to swallow. Analyses of the upper total prosthesis show micromovements ranging from 0.63 ± 0.11 to 1.02 ± 0.13 mm. During simultaneous analyses of the upper prosthesis and lower partial prosthesis, interference was not observed. This pilot study demonstrated the feasibility of using the EMA-3D system to evaluate micromovements in removable prostheses during mastication. Further research is needed to evaluate a larger sample and assess the clinical implications of these micromovements.

## 1. Introduction

Edentulism is defined as a state of oral health that corresponds to the partial or total absence of teeth; it is more prevalent in the elderly worldwide [1,2]. There is a large number of variables associated with tooth loss [3]; the main causes are caries and periodontal disease [4]. Edentulism can generate negative consequences in terms of self-esteem and interpersonal relationships of the elderly [5], in addition to alterations in the functions of phonation and mastication [6]. Therefore, there is a need for the replacement of tooth loss with some type of rehabilitative treatment that allows the restoration of function and aesthetics to the individual [3]. 

The world population growth rates, along with the extended life expectancy, may lead to an increasing demand for oral rehabilitation treatments [7]. Among the treatment possibilities for people with edentulism is rehabilitation with removable prostheses, and their adjustment is based mainly on their ability to resist displacement in the vertical (retention) and horizontal (stability) direction [8]. Many studies have reported that the quality of life of people after being treated with removable prostheses, whether total or partial, increased considerably in terms of improved chewing ability, smile aesthetics, and satisfaction with the state of their mouth after receiving conventional prosthetic treatment [9,10]. Within removable prostheses, there are muco-supported, dento-muco-supported, and implant-supported types; the last of these has been presented as an alternative to overcome the retention problems presented by the first two; however, it has been shown that this type of prosthesis also presents negative aspects, such as pain, marginal bone loss, and associated infections. These factors, added to the cost of treatments, justify the continued utilization of removable prostheses without implant support, the most used alternative [11,12].

With the use of removable prostheses, the masticatory process may present changes such as becoming unilateral, which generates a tipping movement that affects its retention [13]. There are very small movements in removable prostheses, which are often not detected in the clinical examination, which have been called prosthetic micromovements. These can generate discomfort and dissatisfaction on the part of the users [14,15]; therefore, the importance of this field of study is evident. 

To evaluate such movements with objective and quantifiable systems that allow their detection and recognition during the performance of functions such as mastication, different methods have been used, such as a multichannel magnetometer (MoveTrack, Botronics) [14], kinesiograph recording [16], and the application of a digital wave electromagnetic articulography system [15]. Electromagnetic articulography is one of the available methodologies to record masticatory movements. Among EMA systems on the market, we find AG articulographs developed by Carstens Medizinelektronik (Bovenden, Germany). Their latest model is the AG501 3D articulograph. The equipment consists of nine transmitter coils, with each generating alternate electromagnetic fields at different frequencies and 16 small sensors that must be placed on different specific areas of the participant, from which their spatial position can be recorded over real time in the three axes of space [17]. Due to the influence of electromagnetic fields, a small alternating electric current is induced in the sensors, whose parameters are recorded by the articulograph. The induced current intensity varies according to the distance between the receiver coil and the emitter coil. Based on this, the equipment determines the position of the coil within the measurement area. This tool has been used for the evaluation of swallowing patterns, mastication, and phoneme articulation, among others [15,17,18,19]. The aim of this work was to conduct a pilot study for the determination of micromovements in removable prostheses during mastication, in relation to the mandible or maxilla as appropriate, by means of a 3D electromagnetic articulograph (AG501, Carstens Medizinelektronik, Germany). The arrangement of the sensors and the necessary mathematical operations are described. This work represents a pioneering study in this field.

## 2. Methods

### 2.1. Statement of Ethics

A feasibility analysis was performed to evaluate a measurement model. The study was conducted in accordance with the Declaration of Helsinki and approved by the Ethics Committee of Universidad de La Frontera, protocol code No. 121/22, and the date of approval was 8 March 2023. The volunteer gave written informed consent prior to participation, having been informed of the nature of the study.

### 2.2. Participant and Eligibility Criteria

The participant was an adult over 72 years of age and a wearer of an upper removable total prosthesis and lower removable partial prosthesis with a metal base (Kennedy class I, modification 1, well-fitting). The lower prosthesis had a “lingual plate major connector”, and “bar clasps T-shaped” retainers on teeth 3.1 and 4.1 and singular rests on these same teeth.

The subject did not have a pacemaker, implants or prostheses in the head region that could affect the electromagnetic field [20], oral lesions, or signs or symptoms of temporomandibular joint disorders (TMDs) that would hinder the performance of mandibular movements and was not allergic to peanuts. For the identification of these signs and symptoms of TMD, the screening and clinical examination recommended by the American Academy of Orofacial Pain (1993) [21] was applied. A temporomandibular joint diagnosis by a specialist in temporomandibular disorders was not included.

### 2.3. AG501 Electromagnetic Articulograph Description

The AG501 electromagnetic articulograph (EMA, Carstens Medizinelektronik, Germany) has 16 sensors capable of recording the movements produced in the three planes of space; these sensors were located in different points in the mandible, maxilla, and removal prostheses that will be described later. The electromagnetic articulograph has 3 transmission coils that emit electromagnetic fields at different frequencies. These induce electric current in the receptor coils, and the equipment uses this information to determine the position of each receptor coil [17]. This device is certified by the Federal Communications Commission (independent US government agency) as a low-power transmission device. This range is smaller than the frequency range of radio transmission devices like cell phones (10 MHz to 300 GHz) [22].

Articulography devices are safe for health, complying with various standards regarding exposure to magnetic fields [23]. However, there are certain considerations with regard to patients who have implanted devices such as pacemakers [24], cochlear implants [25], or insulin pumps [26] because the electromagnetic field can affect their correct operation. Some studies warn about exposure for pregnant women due to the effects of the electromagnetic field not being clear, which is why it is preferable to avoid risks [27].

During the recordings, there are two types of sensors in relation to their function: the reference sensors, which are used to eliminate the head movements, and the movement sensors, which are used to record the movement. 

Before starting the recordings, a first recording was made as a reference to eliminate the head movements. EMA has a procedure called “Head Correction” for this purpose [28]. This consists of realizing a first recording and using data collected from sensors located in the cutaneous point of the right mastoid, left mastoid, and glabella [17]. 

### 2.4. Recording Method

#### Location of EMA-3D Sensors

The objective was to evaluate the micromovements produced in the upper removable total prosthesis and lower removable partial prosthesis with a metal base during mastication. For this, a total of 12 of the EMA-3D sensors were used. Three of them (1st, 2nd, and 3rd) were located in the accessory of the device called the bite plane (Figure 1), which aligns the horizontal plane of the system with the occlusal plane of the patient. Nine of the EMA-3D sensors were fixed in specific points on the participant using a biocompatible tissue glue (Epiglu^®^, Meyer Haake, Germany), while three of them were used as reference sensors and were located in the cutaneous point of the right mastoid (4th), left mastoid (5th), and glabella (6th), as shown in Figure 2. These sensors overall, with those of the bite plane, were used for the “Head Correction” function of the unit. The “Head Correction” function allows the head movements to be isolated from the mandibular movements. These sensors were present in each recording as a reference of the system, so that only the recording of the mandibular movements was obtained regardless of whether the patient moved their head.
Arrangement of sensors in the upper total removable prosthesis:

Three sensors were placed on the upper prosthesis (7th, 8th, and 9th) and distributed in such a way that they were separated at least 8 mm and did not generate interferences with each other at the time of performing the evaluation (Figure 3).
Arrangement of sensors in the lower removable partial prosthesis:

Two sensors (10th and 11th) were placed on the lower removable partial prosthesis with a metal base and one on the mandibular interincisal midline (12th) (Figure 4).

Once the subject had all 12 sensors attached (three sensors in the bite plane, three reference sensors on the head, three on the upper total prosthesis, two on the lower partial prosthesis, and one in the mandible; Figure 5), the same procedure as in the previous recording was followed.

### 2.5. Recordings

For the evaluations, 3.7 g of peanuts were used [29], which were given to the participant after all the EMA-3D sensors were correctly positioned. The recording of masticatory movements began with the participant in the maximum intercuspation position (MIP) with the test food located between the tongue and the prosthetic palate. The participant was asked to begin chewing freely without indicating a preference side or number of chewing cycles. The recording ended when the first swallow began. This process was repeated ten times with a two-minute pause between each round, with the aim of obtaining ten chewing recordings. During the break, the participant drank water to remove possible residues from the oral cavity. Finally, three recordings were made with the patient in the mandibular postural position to evaluate whether there was interference between the sensors located in the upper total removable prosthesis or lower removable partial prosthesis during rest.

### 2.6. Data Processing

The data were recorded, labeled, and transferred to another computer for processing, for which MATLAB^®^ routines (R2020a, version 9.8.0, The MathWorks Inc., Natick, MA, USA) was used for the position data matrix provided by the AG501 electromagnetic articulograph, and the data required were obtained through specific calculation routines (scripts) especially developed for this study.

During each recording, the Euclidean distance between the prosthesis sensors was calculated using the following equation:$$d=\sqrt{{\left(x_{1}-x_{2}\right)}^{2}+{\left(y_{1}-y_{2}\right)}^{2}+{\left(z_{1}-z_{2}\right)}^{2}}$$

Euclidean distance was calculated point-to-point in each recording. After this, the average of that recording was calculated, and finally, the ten results were averaged. The standard deviation was calculated for each recording and averaged.

The standard deviation of distance between the sensors located on the prosthesis was used as a measure of the level of interference between them. The limit to consider the sensor as performing a stable measurement or the presence of micromovement was set to 0.6 mm since the accuracy of the equipment is 0.3 mm [18] (AG501 Manual). 

Once the stability of the prosthesis sensors had been established, the standard deviation of the distance between the sensors and those located on the subject was calculated. In the case of the sensors of the upper prosthesis, the distance with respect to the reference sensors was measured, while, to evaluate the micromovements of the lower prosthesis, the distance between the sensors located on the prosthesis and the sensors located in the mandible was measured (Figure 4).

## 3. Results

### 3.1. Simultaneous Analysis of Upper Total Prosthesis and Lower Partial Prosthesis with a Metal Base

#### 3.1.1. Interference Analysis of Sensors Located on Upper Total Prosthesis and Lower Partial Prosthesis Simultaneously in the Postural Position

In Table 1, the distance and standard deviation of distance for the rest position are listed. The values correspond to the upper right prosthesis sensor (ur), upper left prosthesis sensor (ul), upper central prosthesis sensor (uc), lower right prosthesis sensor (lr), and lower left prosthesis sensor (ll).

#### 3.1.2. Interference Analysis of Sensors Located on Upper Total Prosthesis and Lower Partial Prosthesis Simultaneously

Table 2 shows the mean distance between sensors located in the in the upper partial prosthesis and between sensors of the lower partial prosthesis with standard deviation of the distance for the ten measurements. The standard deviation average of ten recordings is also shown.

#### 3.1.3. Micromovement Analysis in Upper Total Prosthesis

Table 3 shows the micromovements expressed in millimeters that occurred during peanut chewing in each of the sensors located in the upper total prosthesis in relation to the reference sensors located in the participant’s head. As can be seen in the table, during the chewing of the test food, micromovements were witnessed in the upper prosthesis, which were detected by the three sensors, ranging from 0.63 ± 0.11 to 1.02 ± 0.13 mm.

#### 3.1.4. Micromovement Analysis in Lower Partial Prosthesis

Table 4 shows the micromovements expressed in millimeters that occurred during peanut chewing in each of the sensors located on the lower partial denture in relation to the reference sensor located on the mandible. As can be seen in the table, unlike the upper prosthesis, no micromovements were observed in the lower prosthesis during the chewing of the test food.

#### 3.1.5. Tridimensional Movement

Figure 6 shows movement of sensor in the upper prosthesis (blue) and mandible sensor (orange). Figure 6a shows the up and down movement, 6b shows the back forward movement, and 6c shows the lateral movement. Values were normalized with t-normalization.

## 4. Discussion

Micromovements are known as very small movements of removable prostheses that occur during mastication [15] and are one of the main problems faced by removable prosthesis wearers [30]. The feasibility of electromagnetic articulography to detect these micromovements in total and partial removable prostheses was evaluated, and the results showed that it was possible to record micromovements in both prostheses.

Based on the results obtained, we determined that electromagnetic articulography allows us to detect micromovements in removable total and partial prostheses using the presented technique. The results show that the sensors placed over the prosthesis do not interfere with each other in the rest position (Table 1) or during chewing (Table 2). 

The standard deviation of distances between sensors placed on the upper prosthesis shows the presence of micromovement during mastication (Table 3). On the other hand, the results of lower prosthesis analysis do not show the presence of micromovement according to the established criteria (Table 4). It could be inferred that these results are because the retention of the lower prosthesis, being dento-muco-supported, is greater than that of the upper prosthesis, which is muco-supported. The retention of a prosthesis is the resistance to displacement of the prosthetic base vertically or opposite to its insertion axis. Stability is the quality of a prosthesis of being firm, stable, or constant to resist displacement by horizontal or rotational functional stresses [31]. There are different factors related to the success and satisfaction of users with respect to their removable prostheses. These depend on the number, condition, and alignment of the abutment teeth; the health of the periodontal tissue; the design, support, and material used in the prosthesis; and the different ways to achieve stability and retention (closures versus neuromuscular control), depending on whether the prosthesis is partial or total [32]. Therefore, one of the main objectives when planning a prosthesis should be to achieve adequate retention and stability, thus ensuring the success of the treatment.

In Figure 6, it is possible to observe a certain concordance between jaw movement and upper central sensor displacement. Future morphological analyses of correlations can be carried out between mandibular movement and prostheses displacement to determine the delay between them [33]. In the case of the lower prothesis, results like those shown in Figure 6 are impossible given that micromovements are hidden in the mandibular movement. A possible approximation can be achieved through Procrustes analysis [34]. Fixing a micromovement signal to a mandibular signal requires both to be equal if there are no micromovements, since prothesis movement has the same shape as mandibular movement. In theory, in the presence of micromovements, there will be a difference between adjusted prosthesis movement and mandibular movement.

Few similar studies are found in the scientific literature. Chew et al. [35] analyzed the effect of dental adhesives on the stability of complete maxillary removable prostheses using a kinesiograph. A kinesiograph is a system that records the movement of a magnet attached to the structure of interest. The change in magnetic flow is used to determine the position of the magnet relative to a set of coils. This principle is similar to the one used in EMA. In their study, the displacement in the vertical direction was analyzed. They found that the prosthesis showed an oscillating movement with a range of movement of 0.11 and 0.47 mm. This study analyzes the efficiency of three denture adhesives (Fixodent, Super Polygrip, and Secure) and total maxillary well-fitting dentures and ill-fitting dentures created by trimming the borders. A first recording at the rest position was made for well- and poor-fitting dentures without adhesive. These recordings were used as a baseline to measure denture dislodgment. Once the reference recordings were made, subjects were asked to chew with the left and right side for 20 s with their dentures but without adhesive, with both types of dentures. After this, the upper denture was removed, and adhesive was applied. Three measurements were carried out 1, 3, and 5 hs after adhesive application. Again, this was conducted with well- and poor-fitting dentures. They found that well-fitting dentures had less movement than ill-fitting dentures and that the use of adhesive decreased denture movement for both cases. The improvement in denture fixation was more noticeable in the case of the poor-fitting denture. The Secure adhesive showed the greatest reduction in denture movement. All subjects declared that they were more comfortable with this adhesive. No significant differences were found when the time after adhesive application was analyzed.

Rendell et al. [14] used a multichannel magnetometer tracking system to compare the three-dimensional movement of removable maxillary prostheses with poor and good fixation, based on Kapur’s criteria [36]. This study used a similar device to that of Chew but with the possibility to record three-dimensional movement. A total of 24 subjects, 12 with poor-fitting dentures and 12 with well-fitting dentures, were part of the study. Both groups were asked to chew and swallow dried apricots and peanuts. Also, the denture movement was recorded while the subjects read 10 words aloud. The greatest displacement was observed vertically, in the range of 1–2 mm, while for anteroposterior and lateral movement the movement observed was much less, 0.1–1 mm. They found that there was not a very significant difference in prosthesis movement regardless of whether fixation was good or poor. During chewing sequences, they observed that regardless of the values of movement between groups, there was no significant difference between the standard deviation of movement in each dimension, which was greater for a poor-fitting group than for a well-fitting group, especially for vertical movement. When each dimensional movement was compared between groups, only lateral movement presented significant differences. For swallowing, no significant differences were found for any dimension of movement. This result was repeated for speech analysis. 

Hoke et al. [16] compared maxillary prosthesis movement with and without adhesive using electromagnetic articulography. Three foods were used: carrots, raisins, and processed meat sticks. Each food was tested six times, three with adhesive and three without. In each case, a home position was established, and the subject was asked to maintain the rest position for 5 s. The median of the coordinates for this recording was established as the home position. After this, the subject started to chew until feeling the need to swallow. The Euclidean distance between every coordinate and the home position was calculated. The mean distance, maximum distance, and total traveled distances from the home position were calculated. In their study, a 30% reduction in prosthesis movement was observed for each of these variables. The range of movement observed was between 1 and 3 mm. 

Marin et al. [37] used a kineograph to compare micromovements of the maxillary denture with and without adhesive using bread as a test food. They found a vertical maxillary denture movement of 0.9 (±0.5) in dentures without dental adhesive and 0.7 (±0.3) mm in dentures with dental adhesive. They report that these values correspond to the mean of movement during chewing but did not clarify which reference point was used to measure the movement. 

Compagnoni et al. [38] found a range of vertical movement between 0.5 and 0.8 mm in patients with implant-supported fixed complete dentures in a one-year period after the surgery. They used the same recording protocol as Marin. 

Chew et al. [35] and Hoke et al. [16] did not report the precision of the equipment used, so it is difficult to make comparisons with our measurements since the level of error of these recordings is not known. On the other hand, Rendell et al. [14], Marin et al. [37] and Compagnoni et al. [24] report an accuracy of 0.1 mm in their works. The range of motion reported by Rendell et al. [14] ranges from 0.1 to 2 mm and was calculated as the peak-to-peak value of the signal obtained. Marin et al. [37] and Compagnoni et al. [38] calculated the average movement of dentures. In our case, the range of motion ranges from 0.6 to 1 mm but using the standard deviation of Euclidean distance instead of the average distance in each dimension. This is why direct comparisons may not be conclusive due to the differences described; however, it can be stated that the values obtained from the movement by the two methods are in the same order of magnitude. We consider that, in order to analyze this type of micromovement, it is important for these studies to express the margin of error for each piece of equipment and the method used to quantify the movement in order to generate a consistent reference that allows results to be compared.

In the case of Chew et al. [35], the distance obtained was only in one dimension, vertical, whose range of motion is smaller than that found by Rendell et al. [14] in the same dimension. Chew and Hoke used a reference recording to determine the displacement of dentures, while Rendell, Marin, and Compagnoni did not. Chew reported the change in denture movement relative to the recording without adhesive but did not report the movements of recordings made without adhesive. Rendell used the differences between maximum and minimum displacement as a movement indicator. This may explain the values obtained for vertical displacement, around 3 mm for poor-fitting dentures and 2 mm for well-fitting dentures, since this is the peak-to-peak value. Hoke et al. [16] obtained the Euclidean distance without differentiating the motion in the three dimensions. Their analysis is similar to the present work but directly used the Euclidean distance instead of the standard deviation. They found a reduction of 0.3 mm in displacement for the denture when adhesive was used. This may be explained by the adhesive preventing the prosthesis from lowering from the reference position, reducing the distance as a result. In our case, the criterion used is that of the standard deviation since it gives a better idea of the variability of the position of the prosthesis during mastication. An interesting variable in their study is the total distance traveled since it expresses the total movement by the sensors during the recording. This may be a good indicator of micromovements. If the prosthesis were to show an oscillating movement, as observed by Chew et al. [35], the mean distance would remain constant throughout the recording, which would suggest that the prosthesis does not move. On the other hand, the standard deviation evaluates the range of motion experienced by the prosthesis during the recording. In the cases of Marin and Compagnoni, the reference to determine the denture movement was not clear. It is possible that they used the initial position as a reference. However, their values for denture movement are in the same order as our results.

Based on previous works, we consider that the description of how the movement will be quantified and what is used as a reference should be clear. In the present work, a point was used as a reference, with reference and prothesis sensors, and a recording at the rest position was made to control the presence of interference. This is also useful as a reference since it allows the level of movement to be recorded at the rest position as a baseline. The electromagnetic articulography AG501 manual [28] indicates that the minimum distance that should exist between two sensors so that they do not generate interference with each other is 8 mm. In this study, no sensors presented interference. As previously mentioned, the studies by Chew et al. [35], Rendell et al. [14], Hoke et al. [16], Marin et al. [37], and Compagnoni et al. [38] only performed analysis on upper removable prostheses. So far, no studies have been reported that perform micromovement analysis on lower removable prostheses with electromagnetic articulography, probably due to the difficulty of positioning the sensors. Our study is pioneering in this field since it is the first time that measurements with EMA-3D have been carried out on a prosthesis with a metal base without generating interferences. The results showed that during the chewing tests, no micromovements were generated in the metal-based lower partial denture, which, as mentioned before, was a prosthesis with two free ends, with a lingual plate major connector and bar-clasp T-shaped retainers on teeth 3.1 and 4.1 and singular rests on these same teeth.

We highlight the need to continue these investigations, and subjects who wear removable partial prostheses with different designs should be included since this could be an important element to consider when choosing the design of the prosthesis.

## 5. Conclusions

The recording of micromovements using EMA is possible, and the results obtained are within the same magnitude range as other studies. A major study that evaluates the variability between subjects or different fixation methods is also possible.

The general principles of prosthesis design include providing adequate support, stability, and retention, which determine the success and acceptability of the prosthesis. The results obtained show the presence of micromovements in the upper total prosthesis but not in the lower partial prosthesis. With this important advance in the study of prosthetic micromovements, we took the first step toward other research that can be carried out in this area. In the future, we could consider a study that includes metal-based prostheses with different designs; this would be a great contribution to the clinical area since it would provide new relevant data to consider for the design of the dento-muco-supported prosthesis. 

## Figures and Tables

**Figure 1 bioengineering-11-00229-f001:**
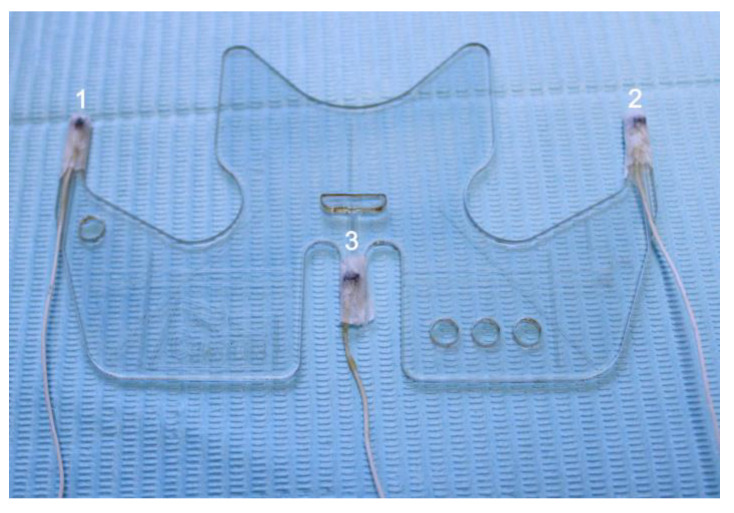
Sensors right (1st), left (2nd), and central (3rd), mounted on the bite plane.

**Figure 2 bioengineering-11-00229-f002:**
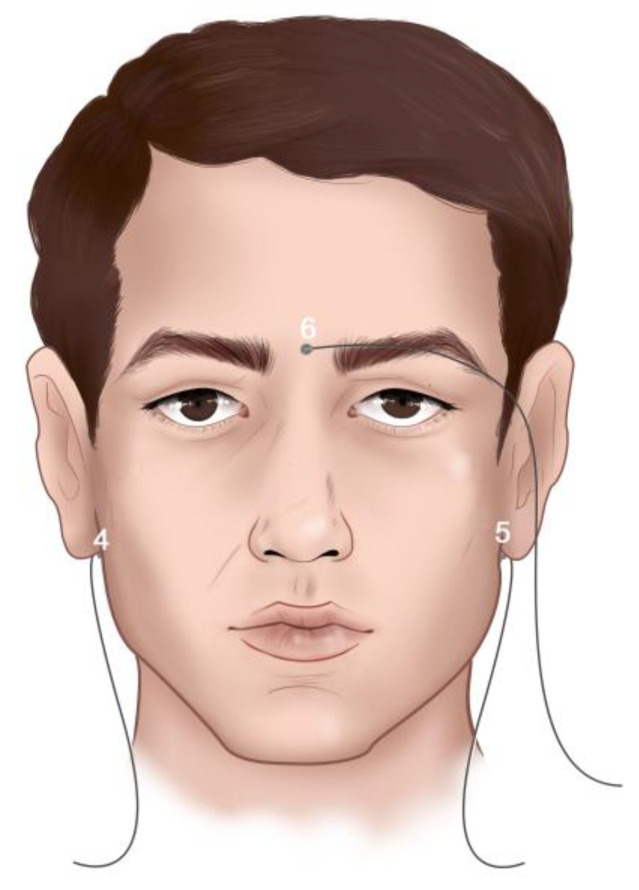
Reference sensors located at three points on the subject’s head: right mastoid skin point (4th), left mastoid skin point (5th), and glabella (6th).

**Figure 3 bioengineering-11-00229-f003:**
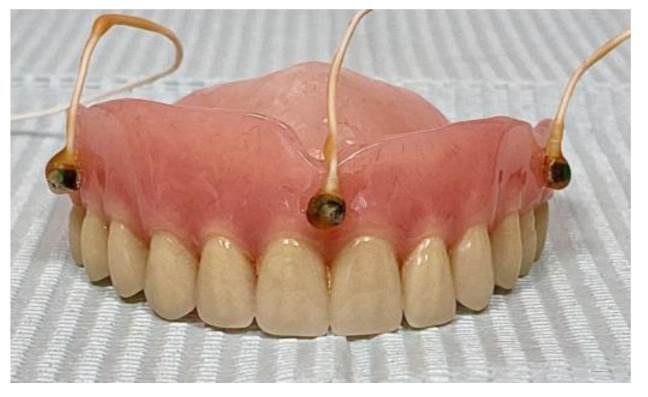
Upper central (uc), upper left (ul), and upper right (ur) sensors (7th, 8th, and 9th), attached to the upper total removable prosthesis.

**Figure 4 bioengineering-11-00229-f004:**
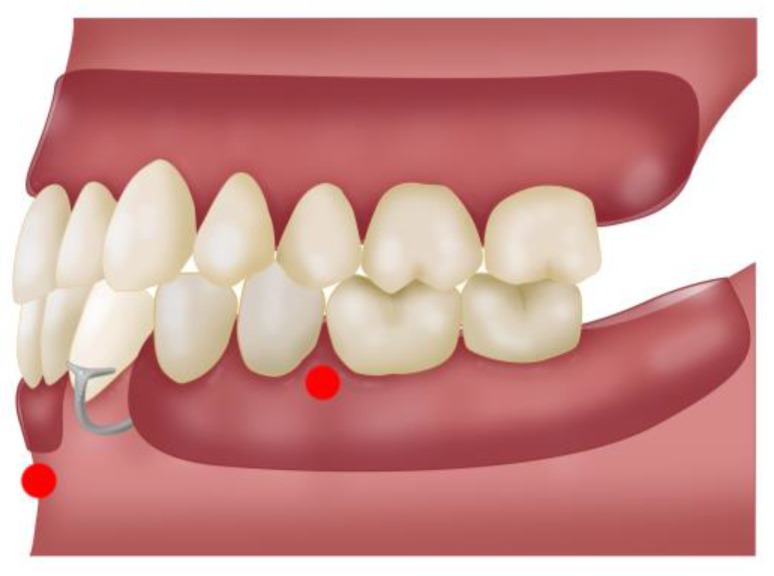
Lower right (lr) and lower left (ll) sensors (10th and 11th) placed in the lower partial removable prosthesis. Central mandibular (cm) sensor, located in the mandibular interincisal midline (12th).

**Figure 5 bioengineering-11-00229-f005:**
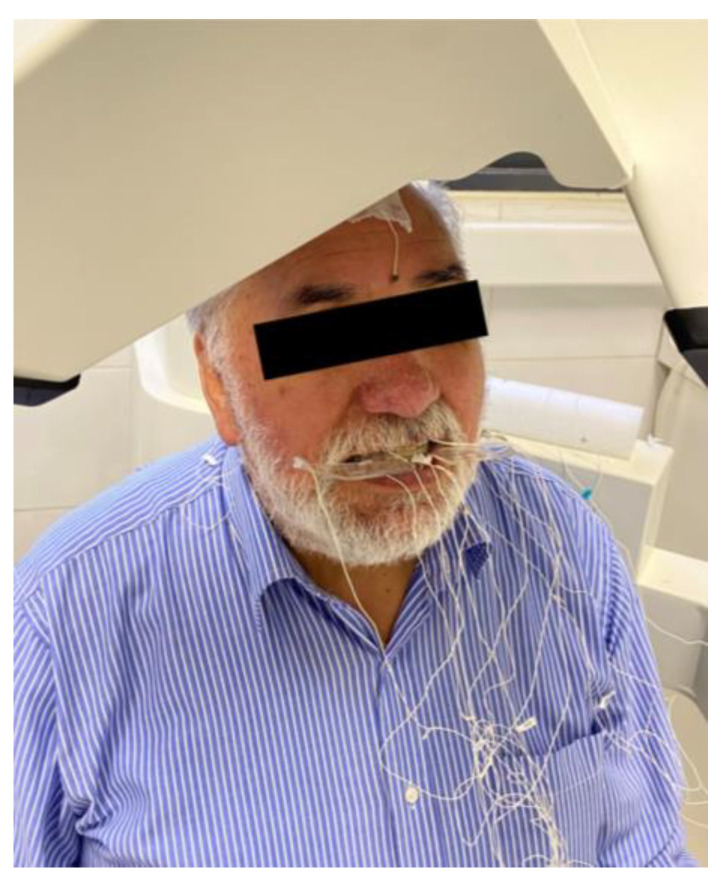
Participant with 12 EMA-3D sensors.

**Figure 6 bioengineering-11-00229-f006:**
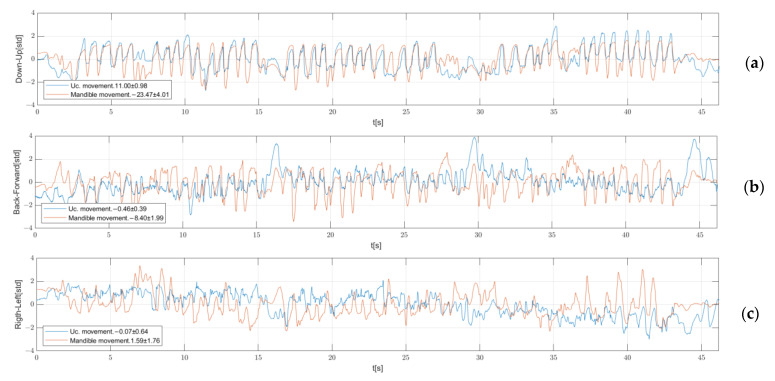
In orange is the mandibular sensor movement, and in blue is the upper central sensor movement. Both graphs are t-normalized. In the legend, the mean and the standard deviation of each one are shown. The Y axis legend indicates the movement direction. (**a**) Vertical movement; (**b**) anteroposterior movement; (**c**) transversal movement.

**Table 1 bioengineering-11-00229-t001:** Distance and standard deviation for the three recordings with the patient in the rest position.

Distances between Pairs of Sensors in Prostheses (mm)			
Sensor 1	Sensor 2	Mean Distance	Standard Deviation
ur	uc	34.93 ± 0.05	0.02 ± 0.00
uc	ul	34.36 ± 0.09	0.02 ± 0.00
ur	ul	52.61 ± 0.03	0.02 ± 0.00
lr	ll	51.98 ± 0.03	0.02 ± 0.00

**Table 2 bioengineering-11-00229-t002:** Distances and standard deviation between all the sensors located in the upper prosthesis, upper right (ur), upper center (uc), upper left (ul), and between the lower right (lr) and lower left (ll) sensors of the lower partial prosthesis.

Distances between Pairs of Sensors in Prostheses (mm)			
Sensor 1	Sensor 2	Mean Distance	Standard Deviation
ur	uc	34.86 ± 0.06	0.03 ± 0.01
uc	ul	34.46 ± 0.04	0.05 ± 0.02
ur	ul	52.62 ± 0.05	0.03 ± 0.01
lr	ll	52.02 ± 0.12	0.05 ± 0.02

**Table 3 bioengineering-11-00229-t003:** Average standard deviation of distance between in the upper prosthesis and reference sensors for the ten recordings. * Indicates that micromovements were recorded.

Micromovement of Upper Prosthesis (mm)	
Sensor	Standard Deviation of Sensor Position
ur	0.74 ± 0.12 *
uc	1.02 ± 0.13 *
ul	0.63 ± 0.11 *

**Table 4 bioengineering-11-00229-t004:** Average standard deviation of distance between the lower partial prosthesis and reference sensor in mandible for the ten recordings.

Micromovement of Upper Prosthesis (mm)	
Sensor	Standard Deviation of Sensor Position
il	0.11 ± 0.03
id	0.12 ± 0.03

## Data Availability

The datasets used and/or analyzed during the current study are available from the corresponding author upon reasonable request.

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
