# Peer review of "Determination of Micromovements in Removable Prosthesis during Mastication: A Pilot Study with 3D Electromagnetic Articulography"

_bioengineering, 2024, doi:10.3390/bioengineering11030229_

Round 1

Reviewer 1 Report (Previous Reviewer 2)

Comments and Suggestions for Authors

Dear Author,

I have thoroughly reviewed your article and I must commend you on the quality of your work. The topic is well-researched, and the presentation of the data is both clear and comprehensive. However, I believe that the article could be further strengthened by including in vitro validation tests.

While your theoretical framework and preliminary data are compelling, in vitro testing would provide essential empirical evidence to support your hypotheses. This would not only reinforce the credibility of your findings but also enhance the overall impact of your research within the scientific community.

I strongly recommend conducting these tests before considering the article for publication. This additional step will likely provide valuable insights and potentially open up new avenues for further research.

Looking forward to seeing these enhancements to your already impressive work.

Best regards,

Author Response

Due to the nature of our study, it is not possible to perform in vitro tests. In this pilot study, all tests were carried out on a volunteer using his prosthesis, since to respond to our study objective, it was necessary for the participant to chew with the prosthesis to detect the presence of micromovements.

Reviewer 2 Report (New Reviewer)

Comments and Suggestions for Authors

Dear Authors, 

The abstract is well written and summary in its various aspects. The keywords are complete and appropriate.

Please check the title. Please check the English style. Please check the references style in the text.

In the introduction:

I suggest the Authors explain in more detail the technology used for this type of study, and how this can actually be useful for diagnostic purposes. I think this type of study is very interesting.

The materials and methods are well described and complete from many aspects.

Results are really well reported.

Discussion: I believe that considering the importance that an examination of this type represents, it would certainly be important in the discussion to also introduce information regarding other works on this topic. The background in the discussion is extremely appreciated by the reader, and gives value to the manuscript itself by improving some aspects.

Conclusions are concise and clear.

Bibliography is relevant and well cited in the text.

Figures and labels are ok. 

Comments on the Quality of English Language

I believe that the style of English as a whole needs to be revised, but the concepts are clear and understandable.

Author Response

Thank you so much for reviewing our manuscript. Your suggestions have been highly helpful to us. You may find our reply (R) below each comment (C).

C1: Please check the title. Please check the English style. Please check the references style in the text.

R1: The English was reviewed and corrected through MDPI's paid editing service.

C2: In the introduction: I suggest the Authors explain in more detail the technology used for this type of study, and how this can actually be useful for diagnostic purposes. I think this type of study is very interesting.

R2: We added this paragraph when we explain in more detail the Electromagnetic Articulograph (EMA) AG501 3D:

“Electromagnetic articulography is one of the available methodologies to record masticatory movements. Among EMA systems on the market, we find AG articulographs developed by Carstens Medizinelektronik (Bovenden, Germany). Electromagnetic articulography is one of the available methodologies to record masticatory movements. Among EMA systems on the market, we find AG articulographs developed by Carstens Medizinelektronik (Bovenden, Germany). Their latest model, the AG501 3D articulograph. The equipment consists of 9 transmitter coils, with each generating alternate electromagnetic fields at different frequencies and 16 small sensors that must be placed on differents specific areas of the participant, from which their spatial position can be recorded over real time in the three axes of space[15]. Due to the influence of electromagnetic fields, a small alternating electric current is induced in the sensors, whose parameters are recorded by the articulograph, Depending on the distance between the receiver coil and the emitter coil, the induced current intensity will vary. Based on this, the equipment determines the position of the coil within the measurement area. This tool has been used for the evaluation of swallowing patterns, mastication, phoneme articulation, among others[13,15–17].”

C3: Discussion: I believe that considering the importance that an examination of this type represents, it would certainly be important in the discussion to also introduce information regarding other works on this topic. The background in the discussion is extremely appreciated by the reader, and gives value to the manuscript itself by improving some aspects.

R3: Thank you for your comment. The next paragraphs were added in order to extend the information of other studies. The added sentences are in bold:

This paragraph extends the description of Chew work (Line 342-361):

Few similar studies are found in the scientific literature. Chew et al. [33] analyzed the effect of dental adhesives on the stability of complete maxillary removable prostheses using a kinesiograph. Kinesiograph is a system that record the movement of a magnet attached to the structure of interest. The change in magnetic flow is used to determine the position of the magnet relative to a set of coils. This principle is similar to the one used by EMA. In their study, the displacement in the vertical direction was analyzed. They found that the prosthesis showed an oscillating movement with a range of movement of 0.11 and 0.47mm. This study analyzes the efficiency of three denture adhesives (Fixodent, Super Polygrip and Secure) and total maxillary well-fitting denture and ill-fitting denture made it by trimming the borders. A first record at rest position was made for well- and poor fitting dentures without adhesive. These records were used as baseline to measure denture dislodgment. Once that the references records were made, subjects were asked to chew with left and right side for 20 seconds with their dentures but without adhesive, with both types of dentures. After this, upper denture was removed, and adhesive was applied. Three measures were made 1, 3 and 5hs after adhesive application. Again, this was made with well and poor-fitting dentures. The found that well-fitting dentures had less movement that ill-fitting denture and that the use of adhesive decreased denture movement for both cases. The improvement on denture fixation was more notorious in the case of poor-fitting denture. The Secure adhesive showed the greatest re-duction of denture movement. All subjects declared that were more comfortable with this adhesive. No significant differences were find when the time after adhesive ap-plication was analyzed.

These paragraphs extend the description of Rendell work (Line 362-378):

Rendell et al.[12] used a multichannel magnetometer tracking system to compare the three-dimensional movement of removable maxillary prostheses with poor and good fixation, based on Kapur's criteria [34]. This study uses a similar device that Chew but with the possibility of record three-dimensional movement. 24 subjects, 12 with poor-fitting dentures and 12 with well-fitting dentures were part of the study. Both groups were asked to chew and swallow dried apricot and peanut. Also, the denture movement was recorded while the subjects read aloud 10 words. The greatest displacement was observed vertically, in the range of 1-2 mm, while for anteroposterior and lateral movement the movement observed was much less, 0.1-1 mm. They found that there was not a very significant difference in prosthesis movement regardless of whether fixation was good or poor. During chewing sequences, they observe that regardless the values of movement between groups was not significant, different between the standard deviation of movement in each dimension was greater for poor-fitting group that for well-fitting group, especially for vertical movement. When each dimensional movement was compared between groups, only lateral movement presented significative differences. For swallowing, no significant differences were found for any dimension of movement. This result repeated it for speech analysis.

This paragraph was added to extend the Hoke work information (Line 379-389):

Hoke et al.[14] compared maxillary prosthesis movement with and without adhesive using electromagnetic articulography. Three foods were used, carrots, raisin and process meat stick. Each food was tested six times, three with adhesive and three without. In each case a home position was stablish asking to subject that maintain rest position for 5 seconds. The median of the coordinates for this record was stablished as the home position. After this, the subject started to chew until felt the need to swallow. The Euclidean distance between every coordinate and the home position was calculated. The mean distance, maximum distance and total traveled distances from home position were calculated. In their study a 30% re-duction in prosthesis movement was observed for each of these variables. The range of movement observed was between 1 and 3 mm.

This paragraph was added to extend the Marin and Compagnoni work information (Line 390-398):

Marin et al. [35] used a kineograph to compare micromovements of maxillary denture with and without adhesive using bread as test food. They found a vertical maxillary denture movement of 0.9 (±0.5) in dentures without dental adhesive and 0.7 (±0.3) mm in dentures with dental adhesive. They report that these values correspond to the mean of movement during chewing but did not clarify which reference point was used to measure the movement.

Compagnoni et al. [36] found a range of vertical movement between 0.5 and 0.8 mm in patients with implant-supported fixed complete dentures in a one-year period after the surgery. The used the same record protocol that Marin.

Reviewer 3 Report (New Reviewer)

Comments and Suggestions for Authors

Dear authors,

Overall, you had a very good idea and it is an interesting research project that could be helpful to other authors. I suggest to address to these comments to improve your manuscript.

·        Please change prosthesess in prosthesis in the title

·        The type of paper submitted is “article” but the paper referred to a single case. So, it is necessary to change the “article” type in “case report”. Thank you.

·        Please change “methods” section in “case presentation”

·        Please explain why do you decided to consider only peanuts. To be rigorous, you have to test some different consistency of food and evaluate the differences regarding denture micromovements while chewing hard or soft food. 

·        Results section: please provide a comment of results. There are any differences between the micromovements of upper or lower prosthesis? Please improve this section with a specific comment for each table to explain your obtained results.

·        Discussion section: it appears very complete but it lacks of limitation section and the comparison between your result with other authors that used similar techniques to evaluate micromovement which you referred on the main text, in order to understand the technique most reliable.

Thank you.

Author Response

Thank you so much for reviewing our manuscript. Your suggestions have been highly helpful to us. You may find our reply (R) below each comment (C).

C1: Please change prosthesess in prosthesis in the title.

R1: We already corrected it. “Determination of micromovements in removable prosthesis during mastication. A pilot study with 3d electromagnetic articulography”.

C2: The type of paper submitted is “article” but the paper referred to a single case. So, it is necessary to change the “article” type in “case report”. Thank you.

R2: We considered that this work corresponding to “article” and no a “case report”, because it is an original research manuscripts, as it reports on new research in this field. A case report “is a detailed report of the symptoms, signs, diagnosis, treatment, and follow-up of an individual patient. Case reports usually describe an unusual or novel occurrence and as such, remain one of the cornerstones of medical progress and provide many new ideas in medicine” (Guidelines To Writing A Clinical Case Report. Heart Views. 2017 Jul-Sep;18(3):104-105. doi: 10.4103/1995-705X.217857. PMID: 29184619; PMCID: PMC5686928). In our case, we dont describe a disease or problem of the patient, our objective is to test a protocol for the evaluation of prosthetic micromovements, but without applying any type of treatment. Our projection is to continue this research with a larger sample.

C3: Please change “methods” section in “case presentation”

R3: As stated in the previous response, we consider our work an “article”, which is why we base ourselves on the instructions given by the magazine for it: “The structure should include an Abstract, Keywords, Introduction, Materials and Methods, Results, Discussion, and Conclusions (optional) sections”.

C4: Please explain why do you decided to consider only peanuts. To be rigorous, you have to test some different consistency of food and evaluate the differences regarding denture micromovements while chewing hard or soft food.

R4: We only used one type of food because this is the first study we carried out in this line of micromovements, so the food chosen was one with which we had already worked before and its behavior was already known to us (Fuentes et al., 2018; Vargas-Agurto et al., 2021; Farfán et al., 2022), our projection is to continue working along these lines with a larger sample of participants, and also including other types of foods, of different consistency and hardness, that are closer to the diet of users who wear prostheses. For this we are already carrying out the corresponding tests, we hope in a not very long period, begin to carry out new evaluations and publish our results.

  • Fuentes, R.; Dias, F.; Álvarez, G.; Lezcano, M.F.; Farfán, C.; Astete, N.; Arias, A. Application of 3D Electromagnetic Articulography in Dentistry: Mastication and Deglutition Analysis. Protocol Report. Int. J. Odontostomat. 2018, 12, 105–112, doi:10.4067/S0718-381X2018000100105.
  • Vargas-Agurto, S.; Lezcano, M.F.; Álvarez, G.; Navarro, P.; Arias, A.; Fuentes, R. Análisis Cinemático de La Masticación de Alimentos Duros y Blandos En Participantes Dentados Utilizando Articulografía Electromagnética 3D. Int. J. Morphol. 2021, 39, 935–940
  • Farfán C, Venegas C, Lezcano MF, Fuentes R. Masticatory function according to body mass index. Part i: Kinematic analysis using different food textures. J. Oral. Res., 11(1):1-11, 2022.

C5: Results section: please provide a comment of results. There are any differences between the micromovements of upper or lower prosthesis? Please improve this section with a specific comment for each table to explain your obtained results.

R5: If there were differences between both prostheses, in the upper prosthesis micromovements were observed, but not in the lower prosthesis, we added this paragraph in table 3: “As can be seen in the table, during the chewing of the test food, micromovements were witnessed in the upper prosthesis, which were detected by the 3 sensors with a ranging from 0.63 ± 0.11 to 1.02 ± 0.13 mm.” And this paragraph in table 4: “As can be seen in the table, unlike the upper prosthesis, no micromovements were ob-served in the lower prosthesis during the chewing of the test food”, to express it.

C6: Discussion section: it appears very complete but it lacks of limitation section and the comparison between your result with other authors that used similar techniques to evaluate micromovement which you referred on the main text, in order to understand the technique most reliable.

R6:

Thank you for your comment, the next paragraphs were added (415-444). The added sentences are in bold:

In the case of Chew et al.[33], the distance obtained is only in one dimension, vertical, whose range of motion is smaller than those found by Rendell et al.[12] in the same dimension. Chew and Hoke used a reference record to determine the displacement of denture while Rendell, Marin and Compagnoni did not. Chew reported the change in denture movement relative to record without adhesive but did not report the movements of records made without adhesive. Rendell used the differences between maxi-mum and minimum displacement as movement indicator.  This may explain the values obtained for vertical displacement, around 3mm for poor-fitting dentures and 2mm for well-fitting dentures, since is the peak-to-peak value. Hoke et al.[14] obtained the Euclidean distance without differentiating the motion in the 3 dimensions. Their analysis is similar to the present work but using directly the Euclidean distance instead the standard deviation of this. They found a reduction of 0.3 mm of displacement for the denture when adhesive is used. This may be explained given that the adhesive prevents the prosthesis from lowering from reference position, reducing the distance from this. In our case, the criterion used is that of the standard deviation, since it gives a better idea of the variability of the position of the prosthesis during mastication. An interesting considered variable in their study is the total distance traveled since ex-press the total movement did it by the sensors during the record. This may be a good indicator of micromovements. If the prosthesis were to show an oscillating movement, as observed by Chew et al. [33], the mean distance would remain constant throughout the recording, which would suggest that the prosthesis does not move. On the other hand, the standard deviation evaluates the range of motion experienced by the prosthesis during the recording. In the cases of Marin and Compagnoni the reference to determine the denture movement is not clear. It is possible that they used the initial position as reference. However, their values for denture movement are in the same or-der that our results.

Based on previous works, we consider that the description of how the movement will be quantified and what is used as a reference should be clear. In the present work a point is used as reference, the reference and prothesis sensors, and a record at rest position is made to control the presence of interference. This also is useful as reference since allows use the level of movement recorded at rest position as baseline.

Reviewer 4 Report (New Reviewer)

Comments and Suggestions for Authors

Dear Authors,

you made a great work! However, some improvements are mandatory before acceptance. 

Comments on the Quality of English Language

A lot of typos and English style throughout the text. 

Author Response

C1: Please check the title.

R1: Was corrected to: “Determination of micromovements in removable prosthesis during mastication. A pilot study with 3d electromagnetic articulography”.

C2: Please check the references style in the text.

R2: It was checked.

C3: In the introduction: Precisely in this regard, it is also interesting to consider the possibilities and negative aspects of a removable prosthetic treatment supported by implants, which can underline in several aspects the advantages of using removable prosthetics without implant support. This could represent a valid alternative to introduce into the evaluation in the introduction. as indicated by: “Reda, R.; Zanza, A.; Di Nardo, D.; Bellanova, V.; Xhajanka, E.; Testarelli, L. Implant Survival Rate and Prosthetic Complications of OT Equator Retained Maxillary Overdenture: A Cohort Study. Prosthesis 2022, 4, 730-738. https://doi.org/10.3390/prosthesis4040057”.

R3: Thank you so much, we added this paragraph considering your suggestion: “Within removable prostheses there are muco-supported, dento-muco-supported, and implant-supported, the latter have been presented as an alternative to overcome the retention problems presented by the first two, however, it has been shown that This type of prosthe-sis also presents negative aspects, such as pain, marginal bone loss, associated infections, among others. These factors, added to the cost of treatments, justify that removable prostheses without implant support continue to be the most used alternative (de Andrade et al., 2015; Reda et al., 2022)”.

C4: In materials and methods section: “not being allergic to peanuts” why? Please explain why peanuts.

R4: We used peanuts because it’s one test food with which we had already worked before and its behavior was already known to us (Fuentes et al., 2018; Vargas-Agurto et al., 2021; Farfán et al., 2022), and this is only a pilot study to prove the protocol of technique, our projection is to continue working along these lines with a larger sample of participants, and also including other types of foods, of different consistency and hardness.

  • Fuentes, R.; Dias, F.; Álvarez, G.; Lezcano, M.F.; Farfán, C.; Astete, N.; Arias, A. Application of 3D Electromagnetic Articulography in Dentistry: Mastication and Deglutition Analysis. Protocol Report. Int. J. Odontostomat. 2018, 12, 105–112, doi:10.4067/S0718-381X2018000100105.
  • Vargas-Agurto, S.; Lezcano, M.F.; Álvarez, G.; Navarro, P.; Arias, A.; Fuentes, R. Análisis Cinemático de La Masticación de Alimentos Duros y Blandos En Participantes Dentados Utilizando Articulografía Electromagnética 3D. Int. J. Morphol. 2021, 39, 935–940
  • Farfán C, Venegas C, Lezcano MF, Fuentes R. Masticatory function according to body mass index. Part i: Kinematic analysis using different food textures. J. Oral. Res., 11(1):1-11, 2022.

C5: Please check the English of the entire manuscript.

R5: The English was reviewed and corrected through MDPI's paid editing service.

Discussion: this section should be complete with the discussion of the outcomes of different papers present in literature. The overall is comprehensive, concise and complete in its various aspects. I think it can be summarized in parts, but it is of a very good standard and offers much of the necessary information.

Thank you for your comment, the next paragraphs were added (415-444). The added sentences are in bold:

In the case of Chew et al.[33], the distance obtained is only in one dimension, vertical, whose range of motion is smaller than those found by Rendell et al.[12] in the same dimension. Chew and Hoke used a reference record to determine the displacement of denture while Rendell, Marin and Compagnoni did not. Chew reported the change in denture movement relative to record without adhesive but did not report the movements of records made without adhesive. Rendell used the differences between maxi-mum and minimum displacement as movement indicator.  This may explain the values obtained for vertical displacement, around 3mm for poor-fitting dentures and 2mm for well-fitting dentures, since is the peak-to-peak value. Hoke et al.[14] obtained the Euclidean distance without differentiating the motion in the 3 dimensions. Their analysis is similar to the present work but using directly the Euclidean distance instead the standard deviation of this. They found a reduction of 0.3 mm of displacement for the denture when adhesive is used. This may be explained given that the adhesive prevents the prosthesis from lowering from reference position, reducing the distance from this. In our case, the criterion used is that of the standard deviation, since it gives a better idea of the variability of the position of the prosthesis during mastication. An interesting considered variable in their study is the total distance traveled since ex-press the total movement did it by the sensors during the record. This may be a good indicator of micromovements. If the prosthesis were to show an oscillating movement, as observed by Chew et al. [33], the mean distance would remain constant throughout the recording, which would suggest that the prosthesis does not move. On the other hand, the standard deviation evaluates the range of motion experienced by the prosthesis during the recording. In the cases of Marin and Compagnoni the reference to determine the denture movement is not clear. It is possible that they used the initial position as reference. However, their values for denture movement are in the same or-der that our results.

Based on previous works, we consider that the description of how the movement will be quantified and what is used as a reference should be clear. In the present work a point is used as reference, the reference and prothesis sensors, and a record at rest position is made to control the presence of interference. This also is useful as reference since allows use the level of movement recorded at rest position as baseline.

Round 2

Reviewer 3 Report (New Reviewer)

Comments and Suggestions for Authors

Thank you. The paper may be accepted in its present form

This manuscript is a resubmission of an earlier submission. The following is a list of the peer review reports and author responses from that submission.

Round 1

Reviewer 1 Report

Comments and Suggestions for Authors

Dear Authors ,

The authors are trying to determine the micro movements in removable prosthesis during mastication with 3d electromagnetic articulography. Although, the study looks interesting, the study design is not strong enough to substantiate the results. The authors have taken two samples and in one sample they are not able to elucidate any results. Stating that EMA is able to record the micro movements leaves an area of uncertainty whether this measurement has been obtained by chance.

In the above scenario, I would suggest the authors should consult a statistician to get a substantial sample size even for pilot study. Later, the results need to be investigated .

With the present sample size and result, I would not suggest publication of the manuscript.

Comments on the Quality of English Language

Minor Spell checks are required 

Reviewer 2 Report

Comments and Suggestions for Authors

The article is well-written, but it would be beneficial to add a section on static tests to assess the error and repeatability of the tool used. Additionally, in the conclusion, it would be helpful to include some information about the clinical significance of the obtained data."

I hope this helps! Let me know if you have other questions or need further assistance.

Comments on the Quality of English Language

minor revision